# Selective Internal Radiotherapy (SIRT) and Chemosaturation Percutaneous Hepatic Perfusion (CS-PHP) for Metastasized Uveal Melanoma: A Retrospective Comparative Study

**DOI:** 10.3390/cancers15204942

**Published:** 2023-10-11

**Authors:** Manuel Kolb, Andrea Forschner, Christoph Artzner, Gerd Grözinger, Ines Said, Helmut Dittmann, Ferdinand Seith

**Affiliations:** 1Department of Diagnostic and Interventional Radiology, University Hospitals Tubingen, 72076 Tübingen, Germany; manuel.kolb@waikatodhb.health.nz (M.K.); christoph.artzner@diak-stuttgart.de (C.A.); gerd.groezinger@med.uni-tuebingen.de (G.G.); 2Department of Radiology, Te Whatu Ora Waikato, Hamilton 3240, New Zealand; 3Department of Dermatology, University Hospitals Tubingen, 72076 Tübingen, Germany; andrea.forschner@med.uni-tuebingen.de; 4Institute of Radiology, Diakonie Klinikum Stuttgart, 70174 Stuttgart, Germany; 5Department of Nuclear Medicine and Clinical Molecular Imaging, University Hospitals Tubingen, 72076 Tübingen, Germany; ines.said@student.uni-tuebingen.de

**Keywords:** uveal melanoma, melphalan, liver, Response Evaluation Criteria in Solid Tumors, selective internal radiotherapy, chemosaturation

## Abstract

**Simple Summary:**

Uveal melanoma usually shows a liver-dominant metastasis spread. Even in the light of multiple treatment options, including liver-targeted therapies, patients die early from this disease. Two promising treatment options are transarterial radioembolization (SIRT) and chemosaturation-percutaneous hepatic perfusion (CS-PHP), and in this study, we retrospectively compared their efficiency on 62 patients (SIRT = 34, CS-PHP = 28) receiving multiple treatment cycles. By a standardized evaluation of the tumor response using RECIST 1.1, we saw a disease control rate of 18% for SIRT and in 30% for CS-PHP. The median of progression-free survival was 127.5 days for SIRT and 408.5 days for CS-PHP, and advanced analysis showed this to be not significant. The median overall survival after treatment was 300.5 days for SIRT and 516 days for CS-PHP, and advanced analysis showed this to be significant. We conclude that for these patients, CS-PHP might therefore be preferable.

**Abstract:**

Even with liver-targeted therapies, uveal melanoma with hepatic metastasis remains a challenge. The aim of this study was to compare the outcome of patients treated with either SIRT or CS-PHP. We included 62 patients with hepatic metastasized uveal melanoma (n = 34 with SIRT, receiving 41 cycles; n = 28 with CS-PHP, receiving 56 cycles) that received their treatments between 12/2013 and 02/2020 at a single center. We evaluated their response according to the RECIST 1.1, as well as progression-free survival (PFS) and overall survival (OS), after the initiation of the first cycle of the liver-directed treatment using Cox regression, adjusted via propensity score analysis for confounders, including the amount of hepatic involvement. The disease control rate was 18% for SIRT and 30% for CS-PHP. The median (range) of PFS was 127.5 (19–1912) days for SIRT and 408.5 (3–1809) days for CS-PHP; adjusted Cox regression showed no significant difference (*p* = 0.090). The median (range) of OS was 300.5 (19–1912) days for SIRT and 516 (5–1836) days for CS-PHP; adjusted Cox regression showed a significant difference (*p* = 0.006). In our patient cohort, patients treated with CS-PHP showed a significantly longer OS than patients treated with SIRT. CS-PHP might therefore be preferable for patients with liver-dominant metastatic uveal melanoma.

## 1. Introduction

Uveal melanoma itself is a rare disease (5.1 per million in the USA), but it is the most common intra-ocular tumor in adults [1]. In the beginning, most patients have no evidence of a metastasized disease [2]. Almost 50% of the patients develop distant metastases during the course of their disease. Different factors influence the likelihood of developing metastatic disease from uveal melanoma. Even in the light of new genetic profiling of cancer subtypes, the size of the primary lesion is considered very important to judge the prognostication of metastatic disease [3]. In about 86% of the patients with metastatic UM, the liver is affected [4]. This liver involvement is prognostic for the patients, as without treatment, the median overall survival (OS) for these patients is only 1.7 months [4]. As of yet, there is no effective option to prevent reoccurrence [5]. In addition, compared to cutaneous melanoma, there are no known options for curative treatment in a metastasized state of uveal melanoma [6]. At the time of diagnosis, only a few patients are eligible for surgical resection [7]. The palliative treatment options are divided into systemic and liver-targeted options. A recent meta-analysis by Khoja L. et al. [8] showed that a median OS for systemic treatment was 9.3 months, whereas for liver-targeted treatment, a median OS of 14.6 months was shown. For some time, the combination of the systemic anti-PD-1 checkpoint inhibitor nivolumab and the systemic CTLA-4 antibody ipilimumab was the most promising treatment option. For this combination, a median OS as high as 19.1 months was reported in a phase II study by Pelster M.S. [9]. Another phase II study by Piulats J.M. [10] reported a median OS of 12.7 months. Recently, a new phase III study showed promising results of Tebentafusp on previously untreated HLA-A*02:01-positive patients with an estimated median OS of 21.7 months [11]. On the other hand, a recent retrospective study showed promising results for the combination of liver-targeted non-endovascular radiotherapy in combination with systemic anti-PD-1 treatment. Especially for the corresponding subgroup of HLA-A*02:01-negative patients, this seems to be very promising [12]. Radiotherapy already has an established and important role in disease control in early metastatic liver diseases [13], as well as primary hepatic malignancies [14].

In most inoperable cases, the whole liver needs to be treated and endovascular options include selective internal radiotherapy (SIRT), using yttrium-90 labeled microspheres and chemosaturation with percutaneous hepatic perfusion (CS-PHP) of melphalan [15,16,17,18,19].

In a recent review of 11 studies, Alexander H. et al. [20] showed a median OS of 12.3 months after SIRT, with a range from 2.8 to 18.6 months. Tulokas S. et al. [15] showed in a retrospective study that the OS after SIRT decreases from 18.7 months if used as a primary therapy to 7.8 months if used as a salvage therapy. 

For CS-PHP, Estler A. et al. [21] found in a retrospective study, a mean OS of 12.9 months. In another retrospective study, Dewald C. [22] showed a mean OS of 12 months after CS-PHP and a non-significantly reduced OS after initial diagnosis when CS-PHP was not a primary therapy. Hughes M. et al. [23] showed an OS of 10.6 months after CS-PHP in a phase III trial, but there was a significant crossover from the control group to the CS-PHP group. Dewald C. et al. [24] showed in a retrospective study of two centers, a median OS of 18.4 months in 66 patients.

Meijer T. et al. [25] showed in a prospective phase II study of CS-PHP, a mean OS of 19.1 months in 35 patients. The highest reported mean OS with 20.0 months in 101 patients was reported in a retrospective study of three centers by Tong T.M.L. et al. [26]. 

Because these treatments require an interdisciplinary team of oncologists, interventional radiologists, nuclear medicine physicians (SIRT) and anesthesiologists (CS-PHP), few centers offer both treatment options and there is a distinct lack of comparison studies of SIRT and CS-PHP [8]. The published data are heterogenous and reported patient cohorts are small, making a direct comparison of both options impossible. In addition, the influence of the individual sites regarding patient selection and the implementation of the different treatments cannot be neglected. 

Thus, the aim of this study was to retrospectively compare the outcome of patients with liver-dominant metastatic uveal melanoma treated with either SIRT or CS-PHP in our department.

## 2. Materials and Methods

The local ethics committee waived informed consent for this single-site retrospective study.

### 2.1. Patient Cohort

We included all adult patients that received either SIRT or CS-PHP in our radiology department with metastasized uveal melanoma between December 2013 and February 2020 following the decision of an interdisciplinary tumor board of our department of dermatology. The follow-up period ended in September 2021, with a recorded date of death or of last recorded contact. Exclusion criteria were: (1) receiving both SIRT and CS-PHP in succession and (2) receiving another liver-targeted therapy (e.g., surgery, interventional ablation or targeted radiotherapy) after the beginning of treatment with SIRT or CS-PHP. A flowchart of the recruiting process is given in Figure 1. We included a total of 62 patients. The patient characteristics, including prior treatments and intra- and extrahepatic tumor load at baseline, are given in Table 1.

### 2.2. Imaging, Response Assessment and Hepatic Tumor Load

For baseline imaging, all patients received both whole-body CT or [18F]-FDG-PET/CT and liver MRI. CT data were acquired with either a Somatom Force or a Definition Flash scanner (Siemens Healthineers, Erlangen, Germany) with a standardized multi-phasic protocol. MRI data were acquired with a 1.5T scanner (MAGNETOM Avanto or MAGNETOM Area, Siemens Healthineers, Erlangen, Germany) or 3T scanner (MAGNETOM Skyra or MAGNETOM Prisma, Siemens Healthineers, Erlangen, Germany) with a standardized protocol consisting of T2w, DWI and multi-phasic T1w dynamic acquisition. All patients received follow-up cross-sectional imaging planned for every three months. In some patients, follow-ups were acquired earlier or later due to changes in health conditions. These off-time follow-ups were included in the analysis. 

The image assessment of all recorded data was performed in a consensus reading by two experienced radiologists (8 and 14 years of experience, respectively). 

Hepatic load in patients receiving CS-PHP was categorized visually into 3 groups of (I) 0–25%, (II) 26–50% and (III) >50% by the radiologist. Hepatic load in patients receiving SIRT was assessed in the pre-treatment Technetium-99m macro-aggregated albumin (MAA) SPECT/CT-scan by an experienced nuclear medicine physician (>25 years of experience) and then re-categorized according to the CS-PHP group. Follow-up imaging was rated in accordance with the Response Evaluation Criteria in Solid Tumors 1.1 (RECIST 1.1) [27] using a dedicated software solution (mint Lesion 3.8.6, Mint Medical GmbH, Heidelberg, Germany). The response evaluation was carried out per treatment cycle as the analysis is dependent on measurable targets and these might change between treatment cycles. 

In addition, depth of response was calculated as the percentual regression of the sum of diameters of RECIST 1.1 target lesions at the timepoint of nadir versus baseline.

### 2.3. Overall Survival and Progression-Free Survival

Both overall survival (OS) and progression-free survival (PFS) were calculated from the time point of (a) initial diagnosis, (b) diagnosis of hepatic metastasis and (c) from the point of first liver-targeted treatment. Calculations of (a) and (b) were made to compare our study cohort to already published data and rule out underlying biases. The analysis of survival times from the point of either SIRT or CS-PHP treatment (c) were made to compare the effectiveness of those therapies. For all patients, time of death or time of last seen alive was recorded. 

In a descriptive analysis of survival times, the true survival times will be between the most restrictive and the most optimistic calculations. For a restrictive assessment, we categorized the date of last seen as the date of death. For a moderate assessment, we only looked at the data of patients with recorded time of death, censoring patients without known death. And lastly, for an optimistic assessment, we categorized all patients without known death as alive until the end of the total observation time (21 September 2021).

As the moderate assessment is considered to be the closest to the true survival, we continued with those estimates to calculate propensity scores (PS) for survival from treatment. PS were calculated in a logistic regression model predicting treatment (SIRT vs. CS-PHP) using the variables age, sex, presence of extrahepatic metastasis and hepatic load at baseline. PS is the resulting predicted probabilities of receiving CS-PHP as a treatment. In addition to age and sex, these variables were selected by consensus between the authors to include the most potential influences on the decision-making process. 

PS was included as a predictor in the regression models along with choice of treatment. The resulting coefficients of treatments indicate the effect of the treatment on the respective dependent variable after summarily controlling for potential confounding influences of the variables used to calculate PS.

For progression-free survival (PFS), there are different approaches as well. PFS is the time until the RECIST 1.1 results in progressive disease (PD). This can be performed for each treatment session or for the whole treatment package. As the aim of the study was to compare SIRT and CS-PHP, and both may be repeated regularly, the evaluation will be restricted on whole treatment, calculating survival from the date of the first treatment session. Similar to the analysis of survival, we calculated progression-free survival in a restrictive, moderate and optimistic version. In the restrictive assessment, patients without PD are considered PD on last RECIST 1.1 observation.

Logistic regression was used for the possibility of calculating depth of response (yes/no). Beta regression was used to calculate the differences in the depth of response. Cox regression was used to calculate the differences in OS and PFS.

### 2.4. Liver-Targeted Treatments

Patients in the SIRT group received Yttrium-90-dotated resin spheres (SIR-Spheres^®^; Sirtex Medical Europe GmbH, Bonn, Germany) as initially published by Kennedy et al. [28] or glass microspheres (Theraspheres^®^; Boston Scientific Medizintechnik GmbH, Düsseldorf, Germany). 

Patients in the CS-PHP group received melphalan using the Hepatic CHEMOSAT^®^ Delivery System (Delcath Systems, Inc., New York, NY, USA) as previously published by Arztner et al. [19] and Vogl et al. [29]. 

Each intervention was performed only after interdisciplinary discussion was conducted individually for each patient. Several treatment sessions of the same kind were summarized as one treatment until a change of treatment was administered.

For SIRT, the treatment can be administered to only parts of the liver such as one lobe or to the entire organ at the same time. A complete treatment of the liver (whole liver at once or serially one side, followed by the other side at an approx. 6 weeks interval) was considered one cycle. Among the 34 patients, 5 received a second additional cycle of treatment with at least a six-month interval from the first. In 2 other patients, the whole liver was serially treated over two interventional sessions (i.e., right and left liver lobes). These 2 patients did not receive an additional cycle. This resulted in a total of 39 cycles over 41 interventional sessions.

For CS-PHP, every interventional session treats the complete liver and is considered one cycle. Fifteen patients received a single cycle of treatment, 7 patients received two consecutive cycles, 4 patients received three consecutive cycles and 3 patients each received four, five and six consecutive cycles, respectively. This resulted in a total of 56 cycles over 56 interventional sessions.

## 3. Results

### 3.1. Tumor Response Evaluation

The hepatic RECIST 1.1 evaluations after each treatment cycle (39 for SIRT and 56 for CS-PHP) showed end results with frequencies as given in Table 2. Three exemplary cases of response evaluation are given in Figure 2.

Extrahepatic progressive disease (PD) can neither be controlled by SIRT nor CS-PHP. After SIRT, 16 (39%) cases of extrahepatic PD were recorded, of which 13 cases were de novo and 5 cases with solely extrahepatic PD. For CS-PHP, there were 14 (25%) cases of extrahepatic PD, of which 9 cases were de novo and no case with solely extrahepatic PD. Evaluating both intra- and extrahepatic disease, RECIST 1.1 evaluations showed an increase in more PD for SIRT, as given in Table 3. The overall disease control rates can be considered 18% for SIRT and 30% for CS-PHP.

Depth of response can only be calculated when a measurable decrease in tumor size is achieved. This was true for 47% of the SIRT cycles and in 52% of the CS-PHP cycles. In a logistic regression model, there was no difference between SIRT and CS-PHP treatment regarding the possibility to calculate depth of response, OR = 1.30, CI 95% (0.48; 3.58), *p* = 0.610. Upon the introduction of PS, there was still no effect of choice of treatment, OR = 1.91, CI 95% (0.60; 6.59), *p* = 0.286.

Referring to the calculated depth of response, the mean and SD for SIRT and CS-PHP were 31.3% ± 26.5% and 35.7% ± 25.2%, respectively. In a beta regression model, there was no difference between SIRT and CS-PHP treatment regarding the calculated depth of response; estimates = 0.19, CI 95% (−0.54; 0.93), *p* = 0.606. Upon the introduction of PS, there was still no effect of choice of treatment; estimates = 0.23, CI 95% (−0.62; 1.10), *p* = 0.596.

### 3.2. Overall Survival

The overall survival calculated in a moderate approach from the diagnosis of disease and from the diagnosis of metastasis are presented in Table 4.

For median overall survival from treatment, SIRT shows 9.90 months and CS-PHP shows 16.96 months, respectively. More detailed results are presented in Table 5. For the differences between restrictive moderate and optimistic calculations, please refer to Appendix A. For the discussion, these results have been recalculated in months. For a better visualization of the overall survival, a Kaplan–Meier graph is given in Figure 3.

In a Cox regression model, there was a significant difference between SIRT and CS-PHP treatment regarding moderately calculated overall survival from treatment, HR = 0.46, CI 95% (0.23; 0.93), *p* = 0.030, without controlling for PS. Upon the introduction of PS, there was still a significant effect of choice of treatment, HR = 0.32, CI 95% (0.14; 0.73), *p* = 0.006. For a better visualization, a survival graph of the Cox regression is given in Figure 4, adjusted for PS.

### 3.3. Progression-Free Survival

For median progression-free survival, SIRT shows 4.16 months and CS-PHP shows 13.43. More detailed results are shown in Table 6. For the differences between restrictive moderate and optimistic calculations, please refer to Appendix A. For the discussion, these results have been recalculated in months.

In a Cox regression model of the moderate calculations, there was no significant difference between SIRT and CS-PHP treatment regarding PFS from treatment, HR = 0.59, CI 95% (0.34; 1.02), *p* = 0.057, without controlling for PS. Upon the introduction of PS, there remained no significant effect of choice of treatment, HR = 0.58, CI 95% (0.31; 1.09), *p* = 0.090.

## 4. Discussion

This is the first comparative study of SIRT and CS-PHP for the liver-targeted therapy of patients with metastasized uveal melanoma. We found a significant longer OS for patients treated with CS-PHP (Table 4 and Table 5). For PFS, there was a trend toward a superior outcome for CS-PHP; however, this was not significant (Table 6). In the evaluation of RECIST 1.1 and depth of response, there was no significant difference of the effectiveness of both therapies. 

Patients with metastasized uveal melanoma have a very limited prognosis. This is mostly due to the liver-dominant metastasizing pattern and the scarcity of effective treatments. In the case of liver metastasis, the OS is limited to 1.7 months without the access to treatments [4]. With the access to treatment, the median OS for those patients rises to 1.07 years as a recent meta-analysis reported [30]. The analysis showed an improved median overall survival for liver-targeted therapies, which is in line with our data. Another meta-analysis showed a median overall survival of 14.6 months for liver-targeted therapies [8]. We found a median overall survival from the diagnosis of liver metastasis of 1.55 years/18.64 months for both therapy options, pooled using a moderated calculation approach. 

As of now, systemic treatments are less successful in uveal melanoma compared to cutaneous melanoma. Yet, there are new systemic therapies like Tebentafusp [31] and a growing awareness of the possible benefits of combination therapies. Combining the presented liver-targeted therapies with immunotherapy appears beneficial [32,33]. Another approach is the combination of systemic immunotherapy with radiotherapy to strengthen the immunostimulatory effects [34]. This seems to be true, especially for patients with limited response to Tebentafusp [12].

We found no significant difference in initial therapeutic effect, e.g., cases that showed measurable tumor reduction (see depth of response). Yet, we saw changes in longer-term outcome variables like overall survival and progression-free survival. For SIRT, we identified a median overall survival of 9.90 months from treatment, 16.83 months from detection of liver metastasis, as well as a median progression-free survival of 4.16 months. For CS-PHP, we identified a median overall survival of 16.96 months from treatment, 23.05 months from detection of liver metastasis, as well as a median progression-free survival of 13.43 months.

When comparing our results to the previously published reports, we see that the current OS is in line with the literature. For SIRT, we have an OS from treatment of 9.90 months compared to 9.69 months by Abbott A.M. [35] and 12.3 months by Alexander H. [20]. For CS-PHP, our OS from treatment of 16.96 months is well in line with the reports of Meijer T. [25] with 19.1 months, Dewald C. [24] with 18.4 months, Hughes M. [23] with 10.6 months, Modi S. [36] with 14.9 months, Brüning R. [37] with 16.7 months and Karydis I. [38] with 15.3 months.

When comparing PFS in our study with the previously reported reports, we find a more limited number of comparable studies, which restricts further investigations. For SIRT, we have a PFS of 4.16 months, whereas Abbott A.M. [35] only found 1.8 months. For CS-PHP, we have a PFS of 13.43 months, which is higher than Meijer T. [25] with 7.6 months, Dewald C. [24] with 8.4 months, Hughes M. [23] with 5.4 months, Modi S. [36] with 8.4 months and Karydis I. [38] with 8.1 months. 

A table comparing all survival data can be referenced in Appendix A.

Our data have to be interpreted within its setting of a retrospective single center study. This was a conscious decision as there is value in reducing the possible bias of technical knowledge between different therapeutic sites when looking at complex procedures such as invasive liver-targeted therapies. In the published works, there is a lack of homogeneous data on those interventional procedures. There is a selection bias in this study as comorbidities of a patient influence the MDT decision and can limit their access to CS-PHP in the case of pre-existing cardiac disease or certain types of vascular shunts. Over the course of our study, the confidence in both treatment modalities has increased and new international research was published especially on CS-PHP, as is the newer of both options. This constant evolution also means a constant change of patient management workflows over time. Up to this moment, no gold standard in the treatment of these patient group has been established. This is reflected in our patient group and no fixed set of rules of treatment can be applied in retrospect. Our approach using propensity scores is aimed at compensating for this bias. And finally, the usage of RECIST 1.1 criteria for the detection of disease progression is limited when multiple treatment cycles are administered, as target lesion can change between the different cycles. On the account of treatment cycles, it is important to note that the majority of our patients received one or two treatment cycles, whereas the current development of CS-PHP treatment shifts to more than three treatment cycles. This shifting approach might explain the higher PFS of CS-PHP in our study as compared to previously reported results. Lastly, for a majority of the patients, the study treatment was not the last treatment to be administered. Even as the differences between the two study groups are small, this could have an influence on the reported OS.

## 5. Conclusions

In conclusion, this study demonstrates a possible small benefit of CS-PHP treatment over SIRT treatment in patients with liver-dominant metastasized uveal melanoma, especially concerning overall survival. Yet, it also shows the eminent need for prospective randomized trials, particularly to find the optimal sequencing of liver-directed therapies, Tebentafusp and immunotherapies or a combination of these therapies.

## Figures and Tables

**Figure 1 cancers-15-04942-f001:**
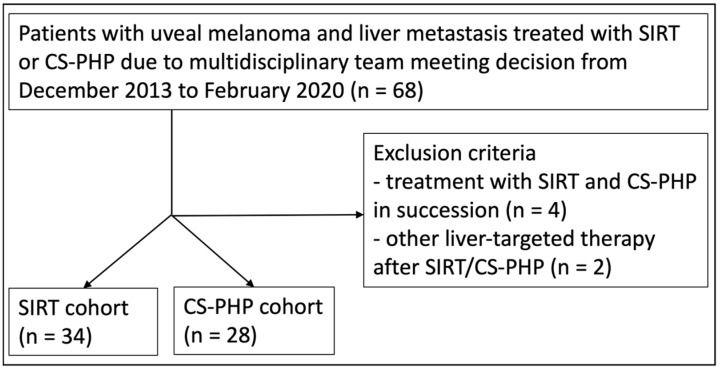
Flow chart of patient selection.

**Figure 2 cancers-15-04942-f002:**
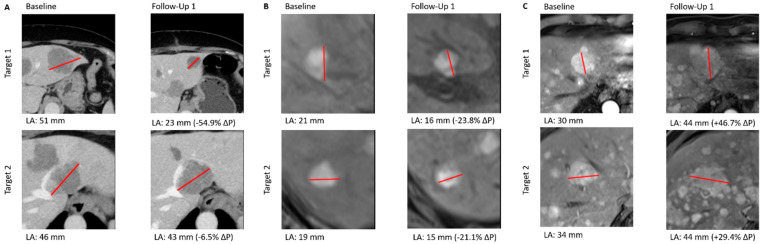
Examples of three different patients from the study population with different responses to study treatments. For each exemplary patient, both liver-targeted lesions according to RECIST 1.1 are given. Red lines indicate the maximum diameter. (**A**): Partial response (PR) after CS-PHP. (**B**): Stable disease (SD) after SIRT. (**C**): Progressive disease (PD) after SIRT.

**Figure 3 cancers-15-04942-f003:**
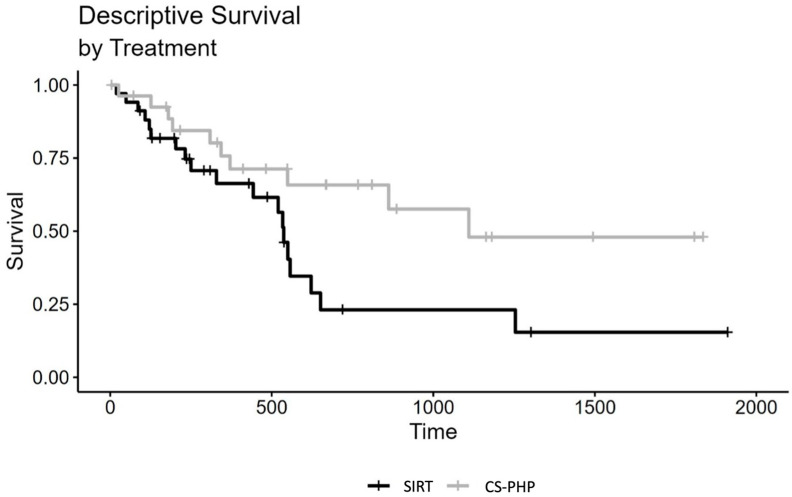
Kaplan–Meier graph of overall survival in days (moderated estimation).

**Figure 4 cancers-15-04942-f004:**
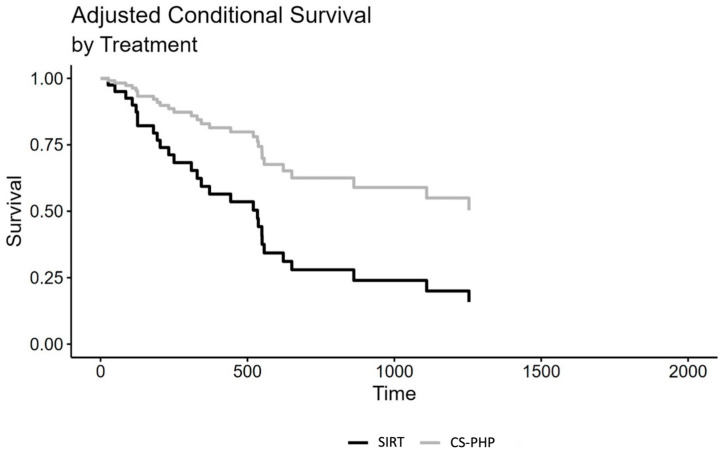
Survival graph of the Cox regression, controlled for propensity scores. Time is given in days.

**Table 1 cancers-15-04942-t001:** Patient characteristics, including treatments.

		SIRT	CS-PHP
Number of patients		34	28
Age in years (mean ± SD)		71 ± 10	63 ± 13
Female sex		15 (44%)	18 (64%)
Pre-treatment	liver-targeted	6 (18%)	6 (21%)
	systemic total	10 (29%)	10 (36%)
	chemotherapy	4 (12%)	1 (4%)
	immunotherapy	6 (18%)	9 (32%)
Extrahepatic metastasis at baseline		14 (41%)	19 (68%)
Hepatic load	0–25%	26 (76%)	16 (57%)
	26–50%	8 (24%)	8 (29%)
	>50%	0	4 (14%)
Total number of cycles of SIRT/CS-PHP		39	56
Post-treatment	systemic total	16 (47%)	20 (71%)
	chemotherapy	2 (6%)	5 (19%)
	immunotherapy	14 (41%)	17 (61%)
	other systemic	0	2 (7%)

**Table 2 cancers-15-04942-t002:** RECIST 1.1 evaluation of hepatic response to study treatment.

Final Hepatic RECIST 1.1 Result	SIRT	CS-PHP
no follow-up	4 (10%)	1 (2%)
complete response (CR)	0	0
partial response (PR)	2 (5%)	5 (9%)
stable disease (SD)	6 (15%)	10 (18%)
non-CR/non-PD	0	1 (2%)
progressive disease (PD)	27 (69%)	39 (70%)

**Table 3 cancers-15-04942-t003:** RECIST 1.1 evaluation of overall response to study treatment.

Final Total RECIST 1.1 Result	SIRT	CS-PHP
no follow-up	4 (10%)	1 (2%)
complete response (CR)	0	0
partial response (PR)	1 (3%)	5 (9%)
stable disease (SD)	2 (5%)	10 (18%)
non-CR/non-PD	0	1 (2%)
progressive disease (PD)	32 (82%)	39 (70%)

**Table 4 cancers-15-04942-t004:** Overall survival of the study population from the timepoint of diagnosis and first detection of first hepatic metastasis.

OS in Days	Estimation		Total	SIRT	CS-PHP
from diagnosis	moderate	mean ± SD	1807.61 ± 1312.06	1762.09 ± 1277.13	1862.89 ± 1374.78
		median (range)	1341 (154–5998)	1378.5 (239–5998)	1341 (154–5044)
from metastasis	moderate	mean ± SD	750.27 ± 696.89	560.59 ± 436.31	980.61 ± 874.20
		median (range)	567 (47–3642)	512 (47–1973)	700.5 (154–3642)

**Table 5 cancers-15-04942-t005:** Overall survival of the study population from the timepoint of study.

OS in Days	Estimation		SIRT	CS-PHP
from treatment	moderate	mean ± SD	409.94 ± 406.96	628.71 ± 512.90
		median (range)	300.5 (19–1912)	516 (5–1836)

**Table 6 cancers-15-04942-t006:** Progression-free survival from the timepoint of study treatment.

PFS in Days	Estimation		SIRT	CS-PHP
from treatment	moderate	mean ± SD	303.88 ± 392.28	527.71 ± 473.43
		median (range)	127.5 (19–1912)	408.5 (3–1809)

## Data Availability

The data presented in this study are available on request from the corresponding author. The data are not publicly available.

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
