# Peer review of "Selective Internal Radiotherapy (SIRT) and Chemosaturation Percutaneous Hepatic Perfusion (CS-PHP) for Metastasized Uveal Melanoma: A Retrospective Comparative Study"

_cancers, 2023, doi:10.3390/cancers15204942_

Round 1

Reviewer 1 Report

In the Introduction the authors have to consider also the radiotherapy as treatment for liver metastases. Therefore, I strongly advise the autors to add a sentence on the role of radiotherapy in liver metastases and to add the following references:

1. Hepatic Radiotherapy to Anti-PD-1 for the treatment of metastatic uveal melanoma patients. Cancers 2023; 15:493;"

2. Project for interventional Oncology LArge-database in liveR Hepatocellular carcinoma – Preliminary CT-based radiomic analysis (POLAR Liver 1.1); European Review for Medical and Pharmacological Sciences 2022; 26(8), pp. 2891-2899

3. Stereotactic radiotherapy for liver oligometastases; Reports of Practical Oncology and Radiotherapy 2022; 27(1), pp. 32-39

Author Response

Dear Editor,

We would like to thank the referees for their expeditious and professional review and

helpful comments. We have answered all comments and modified the manuscript

accordingly, as listed below.

We are looking forward to hearing from you.

Yours sincerely,

Manuel Kolb

Reviewer 1:

In the Introduction the authors have to consider also the radiotherapy as treatment for liver metastases. Therefore, I strongly advise the autors to add a sentence on the role of radiotherapy in liver metastases and to add the following references:

  1. Hepatic Radiotherapy to Anti-PD-1 for the treatment of metastatic uveal melanoma patients. Cancers 2023; 15:493;"

  1. Project for interventional Oncology LArge-database in liveR Hepatocellular carcinoma – Preliminary CT-based radiomic analysis (POLAR Liver 1.1); European Review for Medical and Pharmacological Sciences 2022; 26(8), pp. 2891-2899

  1. Stereotactic radiotherapy for liver oligometastases; Reports of Practical Oncology and Radiotherapy 2022; 27(1), pp. 32-39

We wish to take the opportunity to thank the reviewer for their time. The role of radiotherapy was indeed not well represented in the submitted version. The suggested papers have been added to the introduction. We think that these very recent publications add value to our revised work.

Reviewer 2 Report

The authors present a paper about "Selective Internal Radiotherapy (SIRT) and Chemosaturation Percutaneous Hepatic Perfusion (CS-PHP) for metastasized uveal melanoma: A retrospective comparative study".

The topic is interesting and it totally deserves attention both froma clinical and research point of view.

The introduction provides useful insights however it would be better to further specify the relationshupt between primary lesion diameters and risk of distant metastases (thicker and wider lesions are associated to higher risk).

Unfortunately figure 1, figure 2, figure 3 and figure 4 are not clearly visible in the pdf format please fix this problem in the revised version.

The authors mention in the multidisciplinary asessment the role of several doctors but no mention is given to radiation oncology: is this figure part of the multidisplinary program in your institution? if yes then add this, if not please clarify the reason of this lacking figure.

In the introduction (or in the discussion section) no mention id provided about the role of local therapis combined with immunotherapy (see PMID: 33847208 for a detailed reference) which has become over the last few years one of the treatment options for liver metastases from uveal melanoma.

Were patients treated by radiotherapy to the liver metastases? please provide further details. 

Author Response

Dear Editor,

We would like to thank the referees for their expeditious and professional review and

helpful comments. We have answered all comments and modified the manuscript

accordingly, as listed below.

We are looking forward to hearing from you.

Yours sincerely,

Manuel Kolb

Reviewer 2:

The authors present a paper about "Selective Internal Radiotherapy (SIRT) and Chemosaturation Percutaneous Hepatic Perfusion (CS-PHP) for metastasized uveal melanoma: A retrospective comparative study".

The topic is interesting and it totally deserves attention both froma clinical and research point of view.

The introduction provides useful insights however it would be better to further specify the relationshupt between primary lesion diameters and risk of distant metastases (thicker and wider lesions are associated to higher risk).

Unfortunately figure 1, figure 2, figure 3 and figure 4 are not clearly visible in the pdf format please fix this problem in the revised version.

The authors mention in the multidisciplinary asessment the role of several doctors but no mention is given to radiation oncology: is this figure part of the multidisplinary program in your institution? if yes then add this, if not please clarify the reason of this lacking figure.

In the introduction (or in the discussion section) no mention id provided about the role of local therapis combined with immunotherapy (see PMID: 33847208 for a detailed reference) which has become over the last few years one of the treatment options for liver metastases from uveal melanoma.

Were patients treated by radiotherapy to the liver metastases? please provide further details.

We wish to take the opportunity to thank the reviewer for their time and input. They have identified both content-related as well as formatting weaknesses in our submitted work and we acknowledge these as such.

We have added additional information to the introduction to reflect the importance of the size of the primary lesion for the risk of developing metastatic disease.

We hope that the changes made to figure placement in the provided cancers template are reflected in the automatic pdf generation.

In our department the multidisciplinary team for skin-related malignancies includes radiation oncologists as they are essential for treatment of many different skin-related malignancies. Figure 1 represents the workflow specific to this study which aims to investigate the performance of two other, in this case interventional, liver-targeted therapies. We have made changes to the introduction accordingly. Especially this recent publication PMID 36672442 has shown a potential for patients who will probably benefit less from Tebentafusp. We have added both the reviewer’s suggestion and the later published paper just mentioned to our revised work. Many patients in our care receive radiotherapy for their primary uveal melanoma.

Yet, none of the patients that were selected for this study have received liver-targeted radiotherapy which makes this collective ideal to study the outcome of SIRT vs. CS-PHP.

We hope that the changes made to the manuscript are in line with the reviewer’s expectation. We believe the changes have significantly improved our work and, thus, we are grateful to the reviewer.

Reviewer 3 Report

The manuscript compares two modalities for liver-targeted therapy of metastatic uveal melanoma. Since there is no gold standard for treatment of metastatic uveal melanoma, and each centre has different treatment regimens, it is important to try and compare the available treatments. Therefore, the subject matter is interesting and important, but the paper needs improvements, both in form and methodology.

General:

1.       The authors should state the criteria for picking one treatment over the other. They state that by including the propensity score in the analyses they correct for patient condition and later on, they hint at some restriction for CS-PHP based on patient general health condition. However, it is not clear is the variables used for the propensity score are the same variables used for treatment allocation. This makes is difficult to interpret the data and to state that CS-PHP is better than SIRT (as the authors do in the conclusion).

2.       The authors should consider if it is really useful to present the analyses with restrictive and optimistic approach in the paper, as they do not add much information. If they think they should be reported, I would suggest including them in the supplementals.

3.       I would suggest revising the language, especially syntax and the use of commas in sentences with subordinate clauses.

Abstract:

4.       The percentages of response rate mentioned in the summary and the abstract do not appear anywhere in the text. Adding up the percentages in table 2 or 3 (partial response + complete response) gives different numbers. The authors should also clarify how they define “response rate”.

5.       The last sentence of the abstract may be too strong a statement. The criteria for treatment decision are not reported, which makes it harder for the reader to interpret these results.

Introduction:

6.       The introduction should be re-phrased and made to flow better. In its current form, it reads as segmented and not as a linear story.

7.       The references in the first paragraph should be revised carefully. For example,

a.       Lines 49-50 read “In about 46% of the patients with metastatic UM, only the liver is affected [3, 4].” But this data is not present in either reference 3 or 4.

b.       The ref used for lines 50-51 “Liver involvement is prognostic and without treatment 50 the median overall survival (OS) for these patients is 2 months [5].” Is a very old one (1991). Perhaps a more recent one could be used, such as 10.1001/jamaophthalmol.2018.2466

8.       The second paragraph (lines 64-67) should be rephrased and the two treatments in question should be explained briefly.

9.       The third, fourth and fifth paragraphs (lines 68-80) include a lot of information and numbers, and it is hard for the reader to get a full picture. I suggest including a table in which the studies are compared. Moreover, some of the studies mentioned as single-arm and some are comparative studies but this is not clear from the text.

10.   Line 52 “prohibit recurrence”: please rephrase (e.g. prevent recurrence)

11.   Throughout the text: “liver targeted” needs a hyphen: “liver-targeted”

12.   Line 84: the authors state that “there seems to be a trend towards a more favorable outcome for CS-PHP” but it is not clear what the basis of this statement is.

Methods:

13.   The authors should describe the treatments modalities and cycles in more detail.

a.       They state that there are 62 patients, 34 receive SIRT and 28 receive CS-PHP, and they also report 42 cycles for SIRT and 56 cycles for CS-PHP. However, they do not state how many cycles each patient received. Moreover, if I understand correctly, SIRT is not usually administered in cycles but it can be administered in either one session or 2 sessions (one per lobe): if that is the case, I would advise against calling them cycles but sessions.

b.       Section 2.4 (Liver targeted treatments): here the authors should explain the treatment plan in detail: criteria for inclusion/exclusion, how many cycles/sessions are usually planned, when the treatment is stopped.

14.   Line 104: “ended September 2021”: please rephrase to “ended in September 2021”

15.   Line 106 “after begin”: please rephrase to “after the beginning”

16.   Line 109: “tumor burden” is called “tumor load” in other parts of the text: it would be better to be consistent

17.   Figure 1:

c.       here the “SIRT” is called “TARE”: although they are synonyms, it is better to use the same term in the whole text

d.       MDT should be spelled out

18.   Table 1:

e.       “Female gender”: please check if gender or sex was recorded

f.        In the SIRT group, pre-treatment the numbers don’t add up: “systemic total” = 11, but “chemotherapy” = 4 and “immunotherapy” = 6. Did one patient receive a systemic therapy different from chemotherapy or immunotherapy?

g.       In the SIRT group, post-treatment the numbers don’t add up: “systemic total” = 17, but “chemotherapy” = 2, “immunotherapy” = 14, “other systemic” = 0.

19.   Lines 121 and 122 “state-of-the-art scanners”: it would be better to just name the model of the scanners instead of defining the “state-of-the-art”.

20.   Lines 129-135: in this paragraph it seems that the hepatic load was evaluated differently for CS-PHP and SIRT, but in table 1 only one classification is mentioned.

21.   Lines 147-148: “In a descriptive analysis of survival times the true survival times will be between the most restrictive and the most optimistic calculations and probably close to a moderate calculation.” This sentence is not very clear and would benefit from rephrasing.

22.   Lines 154-161 (propensity score):

h.       It is not clear how the variable to include in the propensity score were chosen: are they the same criteria used to select the treatment?

i.         If I understand correctly, in order to calculate the propensity score, the dependent variable is binary (treatment 1 or treatment 2): therefore, logistic regression should be used to calculate it, not Cox regression.

23.   Line 164: “PD” should be spelled out at first mention.

24.   Lines 165-166 “as for survival in restrictive, moderate and optimistic assessments.” Since this information it present in the next sentence, I would suggest removing it from here.

Results:

25.   Section 3.1 (Tumor response evaluation) presents results of RECIST evaluation after each treatment cycle (Tables 2 and 3). However, this is of limited utility, since patients must have received different numbers of cycles (some may appear more than once). The authors should present the results for each patient as well.

26.   Lines 204-206: “When looking at extrahepatic progressive disease (PD), which can neither be controlled by SIRT or CS-PHP, there were after SIRT 16 (39%) cases of extrahepatic PD of  which 13 cases were de novo and 5 cases with solely extrahepatic PD.”  And lines 208-209: “Combining the results of intra- and extrahepatic RECIST 1.1 evaluations changes the results to having more often PD for SIRT as shown in table 3.” These sentences should be rephrased.

27.   Lines 214-215: “Depth of response could be calculated in 47% of the SIRT cycles and in 52% of the CS-PHP cycles.”

a.       This sentence is a bit unclear. The reader may understand that depth of response was calculated for each cycle, but I think the authors mean to say that 47% of the SIRT cycles and 52% of the CS-PHP cycles could be used for depth of response calculation.

b.       Moreover, I think it would be more useful to use the depth of response per patient, not per cycle.

28.   Lines 219-223: the authors mention that a logistic regression model was used to evaluate if there was a difference between SIRT and CS-PHP in depth of response. However, logistic regression needs a binary dependent variable, and depth of response is a percentage. The authors should revise the statistics.

29.   Figures 3 and 4: CS-PHP is called “chemosat” in the legend. It would be better to be consistent.

30.   Sections 3.2 (Overall survival) and 3.3 (Progression free survival):

a.       When presenting the results of cox regression, in addition to HR and p value, the authors should report 95% confidence intervals as well.

b.       In the tables, the length of follow up is reported in days, but it is reported in months in the discussion. This may create some confusion.

Discussion:

31.   In the discussion, it would be better to specifically refer to figures and tables numbers when discussing the results.

32.   Line 272 “suffering from”: it would be better not to use emotionally-charged language such as “suffering”

33.   Lines 274-275 “In the case of liver metastasis, the OS is limited to 2 months without the access to treatments [5]”. See comment 7b.

34.   Lines 283-286 “As of now, … there might be new options for treatment strategies.” I would advise rephrasing or removing this paragraph.

35.   Line 290 “measurable tumor reduction”: it is unclear is this refers to table 2 and 3 or not.

36.   Lines 298-311: the paragraphs in which the authors compare the findings from the current paper vs the findings from previous studies. It may help clarity to add a table showing previous and current results.

37.   Lines 317-318: “We saw a certain lack of homogeneous data concerning 317 the value of the analyzed therapies ”: this sentence should be rephrased.

38.   Lines 326-327: “This paradigm shift also explains the difference in PFS between the reported 326 PFS of CS-PHP.”: it is unclear what the authors mean by paradigm shift.

The text would benefit from a revision of the English language, especially in terms of syntax. The authors should also try to link together the very short sentences in the introduction.

Author Response

Dear Editor,

We would like to thank the referees for their expeditious and professional review and

helpful comments. We have answered all comments and modified the manuscript

accordingly, as listed below.

We are looking forward to hearing from you.

Yours sincerely,

Manuel Kolb

Reviewer 3:

The manuscript compares two modalities for liver-targeted therapy of metastatic uveal melanoma. Since there is no gold standard for treatment of metastatic uveal melanoma, and each centre has different treatment regimens, it is important to try and compare the available treatments. Therefore, the subject matter is interesting and important, but the paper needs improvements, both in form and methodology.

We thank the reviewer for their significant time and effort on our submitted paper. We agree that there are weaknesses as identified by this reviewer and we will address all of them individually. For this we have copied the remarks here.

General:

  1. The authors should state the criteria for picking one treatment over the other. They state that by including the propensity score in the analyses they correct for patient condition and later on, they hint at some restriction for CS-PHP based on patient general health condition. However, it is not clear is the variables used for the propensity score are the same variables used for treatment allocation. This makes is difficult to interpret the data and to state that CS-PHP is better than SIRT (as the authors do in the conclusion).

We agree that the selection process needs to be explained more clearly. We added more explanations to this to the methods and limitations sections of the revised manuscript. Above that, we are aware of the fact that it is not possible to evaluate all influences on a decision perfectly in retrospect. This is a known bias of retrospective studies. Over the course of the reported time the confidence in these two treatments changed as new experiences were gained and new international research was published on the topic. This constant reevaluation also means a constant change of patient management workflows over time. We chose propensity scores to reflect the most likely factors of influence of the decision-making process in addition to age and sex. We hope that this study will enable a better selection process in the future.

  1. The authors should consider if it is really useful to present the analyses with restrictive and optimistic approach in the paper, as they do not add much information. If they think they should be reported, I would suggest including them in the supplementals.

We agree with the reviewer and we have moved this information to the supplementals (table 7 and 8).

  1. I would suggest revising the language, especially syntax and the use of commas in sentences with subordinate clauses.

We thank the reviewer for this comment. We aimed to improve language in the revised manuscript and hope that the changes are considered sufficient for publication.

Abstract:

  1. The percentages of response rate mentioned in the summary and the abstract do not appear anywhere in the text. Adding up the percentages in table 2 or 3 (partial response + complete response) gives different numbers. The authors should also clarify how they define “response rate”.

We agree that this was not defined properly, and we made appropriate changes to the abstract and results sections. Since the patients are in a palliative situation, even stable disease is to be considered a treatment success. Therefore, the percentages are calculated by subtracting the percentage of progressive disease from 100%. We thus reworded the term “response rate” to “disease control rate” to appropriately reflect CR+PR+SD.

  1. The last sentence of the abstract may be too strong a statement. The criteria for treatment decision are not reported, which makes it harder for the reader to interpret these results.

We agree that the original wording is a rather optimistic/strong one thus changed the conclusion some to “CS-PHP might therefore be preferable for patients with liver-dominant metastatic uveal melanoma”. For an answer on the criteria for treatment decisions please refer to comment 1.

Introduction:

  1. The introduction should be re-phrased and made to flow better. In its current form, it reads as segmented and not as a linear story.

We have revised the manuscript accordingly. We hope that the reviewer finds the changes sufficient for publication.

  1. The references in the first paragraph should be revised carefully. For example,
  2. Lines 49-50 read “In about 46% of the patients with metastatic UM, only the liver is affected [3, 4].” But this data is not present in either reference 3 or 4.
  3. The ref used for lines 50-51 “Liver involvement is prognostic and without treatment 50 the median overall survival (OS) for these patients is 2 months [5].” Is a very old one (1991). Perhaps a more recent one could be used, such as 10.1001/jamaophthalmol.2018.2466

We thank the reviewer for their suggestions, especially for the publication of Lane et al. We changed this section accordingly.

  1. The second paragraph (lines 64-67) should be rephrased and the two treatments in question should be explained briefly.

We have added brief explanations for both Nivolumab and Ipilimumab. In addition, we rewrote this segmented extensively.

  1. The third, fourth and fifth paragraphs (lines 68-80) include a lot of information and numbers, and it is hard for the reader to get a full picture. I suggest including a table in which the studies are compared. Moreover, some of the studies mentioned as single-arm and some are comparative studies but this is not clear from the text.

We have added the missing information to the individual studies. The important information is now presented in the discussion in a more structured manner and we added a new table to the supplement section (table 9)

  1. Line 52 “prohibit recurrence”: please rephrase (e.g. prevent recurrence)

We have changed the phrase accordingly.

  1. Throughout the text: “liver targeted” needs a hyphen: “liver-targeted”

We have revised the manuscript accordingly.

  1. Line 84: the authors state that “there seems to be a trend towards a more favorable outcome for CS-PHP” but it is not clear what the basis of this statement is.

It is correct that the wording of the sentence suggested a statement different to what it aimed for, namely calling out a lack of adequate data. We have rephrased the sentence to reflect this.

Methods:

  1. The authors should describe the treatments modalities and cycles in more detail.
  2. They state that there are 62 patients, 34 receive SIRT and 28 receive CS-PHP, and they also report 42 cycles for SIRT and 56 cycles for CS-PHP. However, they do not state how many cycles each patient received. Moreover, if I understand correctly, SIRT is not usually administered in cycles but it can be administered in either one session or 2 sessions (one per lobe): if that is the case, I would advise against calling them cycles but sessions.
  3. Section 2.4 (Liver targeted treatments): here the authors should explain the treatment plan in detail: criteria for inclusion/exclusion, how many cycles/sessions are usually planned, when the treatment is stopped.

We agree that this was not explained enough and could have caused confusion. For SIRT, the treatment can be administered to only parts of the liver such as one lobe or to the entire organ at the same time. In all patients in our study, the entire organ was treated with SIRT or CS-PHO. A complete treatment of the liver (whole liver at once or serially one side followed by the other side an appr. 6 weeks interval) was considered one cycle. Within the 34 SIRT patients, 5 received a second additional cycle of treatment with at least 6 months interval from the first. In 2 other patients the whole liver was serially treated over two interventional sessions. These 2 patients did not receive an additional cycle. This results in a total of 39 cycles over 41 interventional sessions.

For CS-PHP, every interventional session treats the complete liver and is considered one cycle. 15 patients received a single cycle of treatment, 7 patients two consecutive cycles, 4 patients three consecutive cycles, 1 patient each received four, five and six consecutive cycles, respectively. This results in a total of 56 cycles.

We have added sections to the methods to clarify the questions of the reviewer.

  1. Line 104: “ended September 2021”: please rephrase to “ended in September 2021”

We have revised the manuscript accordingly.

  1. Line 106 “after begin”: please rephrase to “after the beginning”

We have revised the manuscript accordingly.

  1. Line 109: “tumor burden” is called “tumor load” in other parts of the text: it would be better to be consistent

We have revised the manuscript accordingly.

  1. Figure 1:
  2. here the “SIRT” is called “TARE”: although they are synonyms, it is better to use the same term in the whole text
  3. MDT should be spelled out

We agree that this could cause confusion and we have revised the manuscript accordingly.

  1. Table 1:
  2. “Female gender”: please check if gender or sex was recorded
  3. In the SIRT group, pre-treatment the numbers don’t add up: “systemic total” = 11, but “chemotherapy” = 4 and “immunotherapy” = 6. Did one patient receive a systemic therapy different from chemotherapy or immunotherapy?
  4. In the SIRT group, post-treatment the numbers don’t add up: “systemic total” = 17, but “chemotherapy” = 2, “immunotherapy” = 14, “other systemic” = 0.

Thank you for identifying this typo. We have corrected the table accordingly.

  1. Lines 121 and 122 “state-of-the-art scanners”: it would be better to just name the model of the scanners instead of defining the “state-of-the-art”.

Following this point, we have added the scanner models and the vendor.

  1. Lines 129-135: in this paragraph it seems that the hepatic load was evaluated differently for CS-PHP and SIRT, but in table 1 only one classification is mentioned.

We agree that this was confusing. For patients treated with SIRT, the hepatic tumor load can be calculated in % based on the 99Tc-MAA scans prior the intervention. For CS-PHP-patients, this information is not available. Therefore, we re-categorized the hepatic tumor load from the 99Tc-MAA scans according to the 3 groups based on MRI (0-25%, 26-50% and >50%).We have added more information to the revised manuscript.

  1. Lines 147-148: “In a descriptive analysis of survival times the true survival times will be between the most restrictive and the most optimistic calculations and probably close to a moderate calculation.” This sentence is not very clear and would benefit from rephrasing.

We have revised the manuscript accordingly.

  1. Lines 154-161 (propensity score):

  1. It is not clear how the variable to include in the propensity score were chosen: are they the same criteria used to select the treatment?

Please refer to our answer to number 1.

  1. If I understand correctly, in order to calculate the propensity score, the dependent variable is binary (treatment 1 or treatment 2): therefore, logistic regression should be used to calculate it, not Cox regression.

Cox regression deals with time-to-event data, especially where there is censoring. This is the case for this study. For the analysis of this submitted work we are interested in the period of time from treatment until progression/censoring/death events. Logistic regression deals with whether something happens at all and does not account for censoring. Unfortunately, censoring is part of our collected data. This is why we chose Cox regression. We hope that this approach is acceptable for the reviewer.

  1. Line 164: “PD” should be spelled out at first mention.

We have revised the manuscript accordingly.

  1. Lines 165-166 “as for survival in restrictive, moderate and optimistic assessments.” Since this information it present in the next sentence, I would suggest removing it from here.

We have removed this as suggested.

Results:

  1. Section 3.1 (Tumor response evaluation) presents results of RECIST evaluation after each treatment cycle (Tables 2 and 3). However, this is of limited utility, since patients must have received different numbers of cycles (some may appear more than once). The authors should present the results for each patient as well.

The RECIST evaluation was performed to compare the treatment effect on the liver metastases and to provide an impression on the dynamics of the disease also extrahepatic in the different groups. For this purpose, we consider the RECIST evaluation for each treatment cycle to be more meaningful than an end-of treatment assessment. Moreover, RECIST 1.1 is depended on the selection of optimal target lesions. A target lesion for the first treatment cycle might be suboptimal or even not present for continued evaluation in a following treatment cycle. Analysis of the true oncologic dynamics by RECIST 1.1 can be significantly restricted or even impossible in individual cases and we acknowledge this. Yet, RECIST 1.1 is an internationally recognized method when used correctly. Limiting the patient number to only those with continuously measurable target lesions would exclude most of the patients in the current study. This means a trustworthy RECIST 1.1 evaluation over more than one cycle is not feasible. This also reflects the reality of treatment choices as in these patients’ new cycles were only administered after individual MDM consensus. We are aware that this approach will change with gained insight from studies like the FOCUS trial.

  1. Lines 204-206: “When looking at extrahepatic progressive disease (PD), which can neither be controlled by SIRT or CS-PHP, there were after SIRT 16 (39%) cases of extrahepatic PD of which 13 cases were de novo and 5 cases with solely extrahepatic PD.”  And lines 208-209: “Combining the results of intra- and extrahepatic RECIST 1.1 evaluations changes the results to having more often PD for SIRT as shown in table 3.” These sentences should be rephrased.

We rephrased the sentences to improve them.

  1. Lines 214-215: “Depth of response could be calculated in 47% of the SIRT cycles and in 52% of the CS-PHP cycles.”
  2. This sentence is a bit unclear. The reader may understand that depth of response was calculated for each cycle, but I think the authors mean to say that 47% of the SIRT cycles and 52% of the CS-PHP cycles could be used for depth of response calculation.

This sentence was indeed potentially misleading, and we have revised it accordingly.

  1. Moreover, I think it would be more useful to use the depth of response per patient, not per cycle.

As depth of response is based upon RECIST 1.1 measurements, please refer to our answer to number 25.

  1. Lines 219-223: the authors mention that a logistic regression model was used to evaluate if there was a difference between SIRT and CS-PHP in depth of response. However, logistic regression needs a binary dependent variable, and depth of response is a percentage. The authors should revise the statistics.

We reluctantly must disagree as we evaluate “the possibility to calculate depth of response“, not the differences between amounts of depth.

  1. Figures 3 and 4: CS-PHP is called “chemosat” in the legend. It would be better to be consistent.

This was very misleading, and we have revised the figures accordingly.

  1. Sections 3.2 (Overall survival) and 3.3 (Progression free survival):

  1. When presenting the results of cox regression, in addition to HR and p value, the authors should report 95% confidence intervals as well.

We have changed the manuscript accordingly.

  1. In the tables, the length of follow up is reported in days, but it is reported in months in the discussion. This may create some confusion.

We thank the reviewer for this point. We think that it is precise to compare the survival rates of the different treatments in days in the results section. To compare our results with other publications, we now also reported them in months now in results and discussion.

Discussion:

  1. In the discussion, it would be better to specifically refer to figures and tables numbers when discussing the results.

We have added these references to the discussion.

  1. Line 272 “suffering from”: it would be better not to use emotionally-charged language such as “suffering”

We have revised the manuscript accordingly.

  1. Lines 274-275 “In the case of liver metastasis, the OS is limited to 2 months without the access to treatments [5]”. See comment 7b.

We have revised the manuscript accordingly.

  1. Lines 283-286 “As of now, … there might be new options for treatment strategies.” I would advise rephrasing or removing this paragraph.

We have revised the manuscript accordingly.

  1. Line 290 “measurable tumor reduction”: it is unclear is this refers to table 2 and 3 or not.

We have revised the manuscript to decrease confusion.

  1. Lines 298-311: the paragraphs in which the authors compare the findings from the current paper vs the findings from previous studies. It may help clarity to add a table showing previous and current results.

We have added a table to the supplement of the manuscript as suggested.

  1. Lines 317-318: “We saw a certain lack of homogeneous data concerning 317 the value of the analyzed therapies ”: this sentence should be rephrased.

We have revised the manuscript accordingly.

  1. Lines 326-327: “This paradigm shift also explains the difference in PFS between the reported 326 PFS of CS-PHP.”: it is unclear what the authors mean by paradigm shift.

We have revised the manuscript accordingly.

Comments on the Quality of English Language

The text would benefit from a revision of the English language, especially in terms of syntax. The authors should also try to link together the very short sentences in the introduction.

We thank the reviewer for the opportunity to submit a revised manuscript and hope that they find it more suitable for publication.

Round 2

Reviewer 2 Report

The revised version of this article has improved significantly.

I have no further comments.

Author Response

We thank the reviewer for his approval.

Reviewer 3 Report

Review for authors:

The authors have considerably improved the manuscript and have replied to most of the comments. Now the text is easier to read and understand.

However, a few issues remain (the lines refer to the PDF version):

1.      One of the comments of my previous review, was about the propensity score. I still have doubts about that.

a.      The text reads as follows: “As the moderate assessment is considered to be closest to the true survival, we continued with those estimates to calculate propensity scores (PS) for survival from treatment. PS were calculated in a Cox regression model predicting treatment (SIRT vs. CS-PHP) using the variables age, sex, presence of extrahepatic metastasis and hepatic load at baseline. PS are the resulting predicted probabilities of receiving CS-PHP as a treatment. In addition to age and sex, these variables were selected by consensus between the authors to include the most potential influences on the decision-making process. PS were included as a predictor in a logistic regression model along with choice of treatment. The resulting coefficients of treatments indicate the effect of the treatment on the respective dependent variable after summarily controlling for potential confounding influences of the variables used to calculate PS.”

b.      From this text, it seems that the cox regression was used to calculate the propensity score and then the logistic regression was used to study the effect of treatment. Logistic regression may have been used for the “possibility to calculate depth of response” including the PS as a co-variate, but cox regression was used to calculate survival including PS as a co-variate.

c.      From the text it is not clear what type of regression was used to calculate the PS itself: logistic regression or cox regression?

2.      Another one of my previous comments was about the calculation of the depth of response.

a.      The text reads as follows: “Depth of response can only be calculated when measurable decrease of tumor size was achieved. This was true for 47% of the SIRT cycles and in 52% of the CS-PHP cycles. In a logistic regression model there was no difference between SIRT and CS-PHP treatment regarding the possibility to calculate depth of response, OR = 1.20, p = 0.718. Looking at the total number of treatment cycles that allowed for a calculation of depth of response the mean and SD for SIRT and CS-PHP were 31.3% ± 26.5% and 35.7% ± 25.2%, respectively. In a logistic regression model, there was no difference between SIRT and CS-PHP treatment regarding the depth of response, OR = 0.19406, p = 0.606. Upon introduction of PS, there still was no effect of choice of treatment, OR = 0.20802 p = 0.634.”

b.      I understand that the first underlined sentence refers to the possibility to calculate depth of response.

c.      It is not clear if the second underlined sentence refers to the difference in the  possibility to calculate depth of response as well or to the difference in depth of response. If it is the possibility to calculate depth of response, I agree with the logistic regression, but I don’t understand how this calculation is different from the first one.

3.      Results, lines 234-5: “overall disease control rates can be considered 17% for SIRT and 30% for CS-PHP.” I think the 17% should be 18% (looking at the numbers in the table).

4.      When the authors report confidence intervals in the cox regression results, the confidence intervals do not match the hazard rate. Therefore, I think those are not the confidence intervals of the hazard rate, which are the ones that are usually reported for cox regression.

5.      Discussion, line 330: “median progression-free survival of 4.21 months”: elsewhere in the text this number is 4.16

6.      Discussion, line 333: “median progression-free survival of 13.45 months”: elsewhere in the text this number is 13.43

The language has improved.

Author Response

The authors have considerably improved the manuscript and have replied to most of the comments. Now the text is easier to read and understand.

We thank the reviewer for their work and will respond to all points individually.

However, a few issues remain (the lines refer to the PDF version):

  1. One of the comments of my previous review, was about the propensity score. I still have doubts about that.

  1. The text reads as follows: “As the moderate assessment is considered to be closest to the true survival, we continued with those estimates to calculate propensity scores (PS) for survival from treatment. PS were calculated in a Cox regression model predicting treatment (SIRT vs. CS-PHP) using the variables age, sex, presence of extrahepatic metastasis and hepatic load at baseline. PS are the resulting predicted probabilities of receiving CS-PHP as a treatment. In addition to age and sex, these variables were selected by consensus between the authors to include the most potential influences on the decision-making process. PS were included as a predictor in a logistic regression model along with choice of treatment. The resulting coefficients of treatments indicate the effect of the treatment on the respective dependent variable after summarily controlling for potential confounding influences of the variables used to calculate PS.”
  2. From this text, it seems that the cox regression was used to calculate the propensity score and then the logistic regression was used to study the effect of treatment. Logistic regression may have been used for the “possibility to calculate depth of response” including the PS as a co-variate, but cox regression was used to calculate survival including PS as a co-variate.
  3. From the text it is not clear what type of regression was used to calculate the PS itself: logistic regression or cox regression?

We agree that this section needed improvement as there was a mistake inside. The PS were calculated with logistic regression. We have revised the manuscript accordingly.

  1. Another one of my previous comments was about the calculation of the depth of response.
  2. The text reads as follows: “Depth of response can only be calculated when measurable decrease of tumor size was achieved. This was true for 47% of the SIRT cycles and in 52% of the CS-PHP cycles. In a logistic regression model there was no difference between SIRT and CS-PHP treatment regarding the possibility to calculate depth of response, OR = 1.20, p = 0.718. Looking at the total number of treatment cycles that allowed for a calculation of depth of response the mean and SD for SIRT and CS-PHP were 31.3% ± 26.5% and 35.7% ± 25.2%, respectively. In a logistic regression model, there was no difference between SIRT and CS-PHP treatment regarding the depth of response, OR = 0.19406, p = 0.606. Upon introduction of PS, there still was no effect of choice of treatment, OR = 0.20802 p = 0.634.”
  3. I understand that the first underlined sentence refers to the possibility to calculate depth of response.
  4. It is not clear if the second underlined sentence refers to the difference in the possibility to calculate depth of response as well or to the difference in depth of response. If it is the possibility to calculate depth of response, I agree with the logistic regression, but I don’t understand how this calculation is different from the first one.

We thank the reviewer for pointing us towards an unclear paragraph. We have reworded it to reflect that the second part is talking about the actual differences, and we have revised the statistics. The possibility of calculation should be logistic regression as its binary and the differences of depth should be beta regression as the results are in percentages/between 0 and 1. We added this also to the methods chapter.

  1. Results, lines 234-5: “overall disease control rates can be considered 17% for SIRT and 30% for CS-PHP.” I think the 17% should be 18% (looking at the numbers in the table).

We thank the reviewer for noticing this error and have changed the manuscript accordingly.

  1. When the authors report confidence intervals in the cox regression results, the confidence intervals do not match the hazard rate. Therefore, I think those are not the confidence intervals of the hazard rate, which are the ones that are usually reported for cox regression.

The reported confidence intervals were for the coefficient, and we recalculated them (exp(coef)) to make their relation to the HR more obvious and to avoid confusion we have removed the coefficient.

  1. Discussion, line 330: “median progression-free survival of 4.21 months”: elsewhere in the text this number is 4.16

We thank the reviewer for noticing this error and have changed the manuscript accordingly.

  1. Discussion, line 333: “median progression-free survival of 13.45 months”: elsewhere in the text this number is 13.43

We thank the reviewer for noticing this error and have changed the manuscript accordingly.

Comments on the Quality of English Language

The language has improved.

We thank the reviewer for their approval.